# Visual Evoked Potentials as Potential Biomarkers of Visual Function in Patients with Primary Sjögren’s Syndrome

**DOI:** 10.3390/jcm10184196

**Published:** 2021-09-16

**Authors:** Edyta Dziadkowiak, Agata Sebastian, Malgorzata Wieczorek, Anna Pokryszko-Dragan, Marta Madej, Marta Waliszewska-Prosół, Sławomir Budrewicz, Piotr Wiland, Maria Ejma

**Affiliations:** 1Department of Neurology, Medical University of Wroclaw, 50-556 Wroclaw, Poland; anna.pokryszko-dragan@umed.wroc.pl (A.P.-D.); marta.waliszewska-prosol@umed.wroc.pl (M.W.-P.); slawomir.budrewicz@umed.wroc.pl (S.B.); maria.ejma@umed.wroc.pl (M.E.); 2Department of Rheumatology and Internal Medicine, Medical University of Wroclaw, 50-556 Wroclaw, Poland; agata.sebastian@umed.wroc.pl (A.S.); marta.madej@umed.wroc.pl (M.M.); piotr.wiland@umed.wroc.pl (P.W.); 3Department of Geography and Regional Development, University of Wrocław, 50-137 Wroclaw, Poland; malgorzata.wieczorek@uwr.edu.pl

**Keywords:** primary Sjögren’s Syndrome, aching joints, anti-Ro52 antibodies, visual evoked potentials, hyperexcitability of cortex

## Abstract

Visual evoked potentials (VEP) are changes in potentials that arise in the central nervous system. In the interpretation of the VEP test results, it is assumed that the elongation of the latency time is caused by the demyelination of the nerve fibers, and the axon damage is responsible for the decrease in the amplitude. The observed VEP deviations are not specific for specific diseases, but indicate disturbances in visual conductivity. VEP may play a diagnostic role in the early detection of visual involvement. The aim of the study was the functioning of visual pathway assessment on the basis of visual evoked potentials (VEP) examination, in patients with primary Sjögren’s Syndrome (pSS), without focal symptoms of central nervous system disorder. The effect of disease activity, as assessed by clinical parameters and antibody levels (anti-Ro52, SSA, and SSB), on the central nervous system was also evaluated. Thirty-two consecutive patient with pSS (31 females, 1 male) were included in the study. VEP was performed at baseline, and after 6 (T6) years. Their results were compared longitudinally between the baseline and T6, depending on the duration of the disease and treatment. The immunological activity of pSS was also analyzed. The group of patients showed a significant prolongation of the P100 implicit time (105.5 ± 5.1 vs. 100.6 ± 3.9; *p* = 0.000) and a significant higher the P100-N145 amplitude (12.3 ± 4.1 vs. 9.4 ± 3.0; *p* = 0.000). Abnormalities in electrophysiological parameters of VEP at baseline correlated with presentation of anti-Ro52 antibodies and aching joints. At baseline, the P100 implicit time was shorter for the patients with pSS than for those at T6 (105.50 ± 5.1 vs. 109.37 ± 5.67; *p* = 0.002). pSS patients without CNS involvement presented with dysfunction of visual pathway, as revealed by VEP abnormalities. Relationships were found between VEP parameters and with present of anti-Ro52 antibodies and aching joints. VEP may be a useful method for assessment and monitoring of subclinical visual deficit in the course of pSS.

## 1. Introduction

Primary Sjögren’s Syndrome (pSS) is a chronic, systemic autoimmune disease that occurs in 2–3% of the adult population [1,2,3,4]. The pSS pathomechanism is based on humoral and cellular response disorders, therefore the symptoms spectrum may be broad and include organic disorders such as muscles, eyes, and the nervous system. As suggested in the available reference literature, patients with pSS the damage to the sight organ and/or the visual pathway may anticipate axial pSS symptoms. Additionally, it may appear in cases with established diagnoses and a longer course of the disease [5,6].

In the course of autoimmune connective tissue diseases, the inflammation process may concern each part of the eyeball structure. In these disorders, the frequency and degree of lesions in the eyeball, along with the the prognosis, are variable. They depend on many factors, such as the type of the basic illness, the progression of the inflammation process, and the sex of the patient. Such eye pathologies as xerophthalmia, conjunctivitis, keratitis, episcleritis, scleritis, vasculitis, retinopathy, optic neuritis, and glaucoma and cataract induced by steroid therapy are listed as the most common eye pathologies. In pSS, compared with other autoimmune diseases, optic nerve neuropathy is a rare neurological manifestation, although its exact frequency is unknown. Mostly, it appears in the context of neuromyelitis optica spectrum disorders (NMOSD) [7,8]. Gono et al. [9] found features of nerve II neuropathy in 18% of 32 patients with pSS, and a similar frequency (16%) of optic neuritis was reported by Delalande et al. [10] in a more representative group of pSS patients. However, sight disorders may also be the consequence of ischemic and/or demyelinative lesions to the farther parts of the visual pathway [11].

Furthermore, eye disorders in patients with pSS may also occur in the course of chloroquine treatment. Side effects of chloroquine include: corneal disorders (swelling, pinpoint or linear opacities, decreased sensitivity to stimuli, deposits in the cornea, blurred vision, halos around light sources, or photophobia), retinal disorders (edema, atrophy, pigmentation disorders macula and the rest of the retina, changes in arterioles, or retinopathy), visual field disturbance, and partial or complete loss of vision [12,13].

The examination of multimodal evoked potentials belongs to non-invasive, sensitive, and repetitive nervous system functioning assessment methods. The analysis enables the detection of even subclinical disorders, in the case of visual evoked potentials (VEP), in the optic pathway. Considering clinical significance of a visual deficit which may seriously affect the patients’ functioning, it is important to identify, as early possible, those pSS subjects with a greater risk of visual pathway involvement for their closer follow-up and appropriate therapeutic approach.

The aim of the study was the assessment of the functioning of visual pathway based on VEP examination, in patients with pSS, without focal symptoms of central nervous system disorder or primary disorders in the basic ophthalmological examination. The authors analyzed the relevance between the clinical and immunological activity of pSS and the parameters of VEP.

## 2. Materials

Thirty-one females and one man represented the group of patients, on average 51.03 years of age (51.00 for females and 52.00 for men), who, at the moment of VEP examination, underwent the pSS criteria according to American–European classification from 2002 to 2016 [14,15]. The markers of exclusion included resolved neurological and ophthalmological diseases (except dry eye), metabolic, systemic, deficiency diseases, and using drugs or stimulants that may influence the nervous system’s functioning and the sight organ (except for chloroquine and hydroxychloroquine). The maximal vision defect in patients diagnosed did not go beyond 0.8 diopters.

The control group consisted of 50 healthy volunteers (43 females and 7 males, on average 46.24 years of age; 46.53 for females and 44.43 for males), without any chronic diseases or neurological and rheumatological symptoms, selected from the students and hospital staff with respect to the age and sex. The detailed neurologic examination of the controls was normal.

Each person from the patient group and the control group was informed in detail about the aim and the procedure of the examination and expressed conscious consent in writing for participating in this examination.

## 3. Methods

In every patient, a neurological examination was performed. Patients with neurological symptoms were excluded. In a head scan examination (CT or MRI), an expansive intracerebral process and structural/focal lesions were excluded. Fatigue was assessed using the Functional Assessment of Chronic Illness Therapy (FACIT) and visual analog scale (VAS, 0–10 pts, where 0 denoted total lack of symptom, while 10 referred to intensity affecting everyday home activity). The analysis of pSS activity was conducted using EULAR Sjögren’s Syndrome Disease Activity Index (ESSDAI) [16], EULAR Sjögren’s Syndrome Patients Reported Index (ESSPRI) [17], as well as focus score (FS) in minor salivary glands biopsied from the lower lip and laboratory parameters, including C3 and C4 component levels, anti-nuclear antibodies (ANA), and extractable nuclear antigen panel (ENA) measured by indirect immunofluorescence (IFA), and enzyme-linked immunosorbent assays (ELISA), rheumatoid factor (RF), erythrocyte sedimentation rate (ESR), C-reactive protein (CRP), peripheral blood morphology, and total protein level in serum. The anti-Ro52, anti-SSA and anti-SSB antibodies were determined by ELISA method. The anti-Ro52 level was assessed relying on the intensity of the color directly linked to the number of antibodies in the patient sample (0—no antibodies, 3—the most intense color).

### Visual Evoked Potentials

The evoked potential examination was conducted using the Viking Quest equipment. The procedure was conducted according to the International Federation of Clinical Neurophysiology (IFCN) and American Society of Electroencephalography [18,19] guidelines. The evoked potentials were recorded at a fixed time (after breakfast, before noon). The study was conducted in a sedentary position, in a quiet and dimmed room at 22–24 degrees Celsius. Superficial Ag/AgCl electrodes with a diameter of 10 mm by Nicolet Biomedical Div of Viasys Healthcare, Madison, USA were used and placed on the skin of the head according to the international 10–20 scheme, and fixed using the adhesive-conductive Ten20 Conductive paste by D.O. Weaver and Co. (Aurora, CO, USA)

Eye potentials were evoked using the structural stimulus of a checker-board with alternately changing white and black fields, emitted by the television screen by Nicolet company, model NIC-1005, from the distance of 1 m. The angular magnitude of particular squares made 1.1 degrees, and the whole sight field made 18 × 22 degrees. Consecutively, the left and the right eye was stimulated with 1.88 Hz frequency. The receiving electrode was placed in the central line, in the occipital zone (Oz), the interference electrode was placed in the frontal zone (Fz), and the grounding electrode on the forearm. A total of 75 responses were averaged in the frequency band 1–30 Hz, and the analysis time was 500 ms. Latency wave N75, P100-N145, and N145 were assessed, and inter-ocular difference of latency wave P100 (relative latency) and P100-N145 amplitude complex. In patients with a minor sight defect, the examination was performed in correction glasses.

At baseline, all the patients with pSS and healthy controls had VEP performed. Data on neurological symptoms and rheumatological characteristics over the course of the disease, as well as autoimmune markers, were also collected. During the follow-up visit after 6 years, VEP were repeated. Their results were compared longitudinally between the baseline and T6, depending on the duration of the disease and treatment. Regular ophthalmologic examination included assessment of visual acuity on Snellen’s table, color vision test on Ishihara’s table, and fundoscopy, as recommended in the follow-up for potential HCQ toxicity.

## 4. Statistical Methods

Statistical analysis was conducted using STATISTICA 12.0 software. All tests (for normality, homogeneity of variance, the equivalence of means, and ranked tests) were conducted at the significance level of α = 0.05. To test the normality of distribution, the Shapiro–Wilk test was used. Variables with normal distribution were tested for homogeneity of variance. After positive verification of both hypotheses (of normal distribution and homogenous variance), the hypothesis of equivalent means between both groups was tested using Student’s *t*-test. Comparison of variables, the distribution of which was not normal according to the Shapiro–Wilk test, was conducted using the ranked Mann–Whitney U test. The range of correct VEP parameters was calculated based on average values received in the control group, including the triple standard deviation (x ± 3 SD).

## 5. Results

### 5.1. Rheumatologic Parameters Analysis

In the study group, the average period from the moment of pSS diagnosis to the moment of carrying the VEP examination lasted four years (range 1–14 years). The first symptoms of pSS observed in patients were: xerophthalmia/xerostomia (63%), fatigue (24%), arthralgia lasting for more than 30 min a day (21%), skin lesions characteristic for pSS (12%), pe-ripheral arthritis (9%), and swelling of major salivary glands (6%). At the moment of examination, the symptoms of arthritis were diagnosed in 28 (87%) patients, respiratory system involvement in 19 (59%), swelling of salivary glands (parotid and submandibular glands) in 14 (44%), skin lesions typical for pSS in 10 (31%), and lymphadenopathy in 5 (15%). Skin lesions were found in 10 of patients, of which 6 had a palpable pulpura (i.e., lesions of the vasculitis type) located on the lower limbs. In 96% of patients, the Schirmer’s test value was less than or equal to 5 mm/5 min. The Schirmer’s test was not performed on the same day as the VEP.

As to the coexistent ailments, in one patient, lipid disorder was diagnosed; in one, cholelithiasis; in three, controlled hypertension; and in none, thyroid nodules, without the hormonal disorder.

The average FS in the study group was 2.1 (min 0–max 5) and ESSDAI 20 points (min 4–max 24), which indicated high activity of the disease [15]. In all patients, mean xerophthalmia persisted, expressed in an average ESSPRI scale of 5.2 pts (min 2–max 8). In 31 (94%) patients, mean xerostomia persisted, expressed in an average ESSPRI scale of 5.2 pts (min 0–max 9).

ANA antibodies ≥ 1:320 were present in 26 (81%) patients with pSS; anti-SSA in 26 (81%); anti-Ro52 in 22 (69%); and anti-SSB in 21 (66%). In three cases (9%), the presence of RF was not confirmed. In all patients, the presence of cryoglobulins was not confirmed.

No correlation between pSS activity expressed on the ESSDAI index and the disease duration was found (*p* = 0.18). The baseline characteristics of patients with pSS are presented in Table 1.

In pSS therapy, in 80% of patients, glucocorticosteroids were used in the average dose of 7.5 mg per twenty-four hours in prednisone conversion. In 45% of patients, methotrexate was used orally in the average dose of 15 mg once a week, with respective folic acid supplementation. In 30% of patients, non-steroidal anti-inflammatory medications were used. In 20% of patients, azathioprine was used in the average dose of 125 mg per day and 12% cyclosporine 150 mg per day on average. In the study group, 60% of patients were treated with hydroxychloroquine in the average dose of 200 mg per day, and 25% with chloroquine in the dose of 250 mg per twenty-four hours. All these patients underwent an ophthalmological examination regarding retina tests to exclude contraindications for treating these medications. The examination was carried out before the inclusion of the treatment and every 6 months during the treatment. No symptomatic changes in rearrangements of pigment on the retina were confirmed in any patient.

### 5.2. VEP Parameters

In the study group with pSS, symptomatic differences in P100 implicit time and P100–N145 amplitude in comparison with the control group were demonstrated (Table 2).

Mean P100 implicit time was relatively longer (*p* < 0.001) and mean amplitude relatively higher (*p* < 0.001).

Pathologic VEP was registered in seven patients. In five cases, prolonged P100 implicit time (max 118 ms) was confirmed (one-sided in one case). In three patients, abnormally high P100-N145 amplitudes (max 22.6 µV) were obtained; in one of them, longer P100 implicit time was also registered (Figure 1).

#### 5.2.1. Baseline Analysis of VEP Parameters with Clinical Data

In patients in whom antibodies Ro52 were present, symptomatically prolonged P100 implicit time (*p* = 0.02) was confirmed. The relation of these two factors was presented in Figure 2.

The patients with aching joints showed symptomatically shorter P100 implicit time (*p* = 0.03) in comparison to the study group without this symptom.

Statistical significance between VEP parameters and the degree of xerophthalmia, the disease’s duration, and the presence of skin lesions characteristic for vasculitis, and lesions in major salivary glands was not proven. Neither the correlation between VEP parameters and high ESR, the presence of anti-SSA and -SSB antibodies, or the low complement component C3 and C4 were confirmed.

#### 5.2.2. Analysis of VEP Parameters after 6 Years of Follow-Up

At baseline, the P100 implicit time was shorter for the patients with pSS than for those at T6 (105.50 ± 5.1 vs. 109.37 ± 5.67; *p* = 0.002) (Table 3). At the follow-up visit after 6 years, 31 patients had a longer P100 implicit time, only 1 patient had a shorter latency. While the P100–N145 amplitude decreased in 31 patients, in only one patient, it increased (Table 4). Figure 3 shows an example of a VEP test result in a pSS patient at the baseline and after 6 years. Apart from a positive correlation between disease duration and latency N145 (0.39), no significant correlations of VEP parameters and duration of the disease and duration of treatment were found in longitudinal assessment.

We also analyzed the statistical relationship between VEP parameters and disease duration and treatment time, both at baseline and after 6 years of follow-up. In the group of patients with ≤10 years of disease, the P100 implicit time was 105.18 ± 5.35, while in the group of patients with disease duration of >10 years, the P100 implicit time was 106.00 ± 4.90; *p* = 0.658. After 6 years, the group of patients with shorter disease duration had the P100 implicit time 108.38 ± 5.67, while the group with longer disease duration had the P100 implicit time 110.81 ± 5.81; *p* = 0.249. When the groups were divided according to the duration of ≤5 years of treatment with hydroxychloroquine or chloroquine, the P100 implicit time was found to be 104.89 ± 5.20, and 106.23 ± 5.09 in the group of patients treated for more than 5 years; *p* = 0.461. After 6 years, the P100 implicit time was found to be 109.01 ± 6.63 in the group of shorter-treated patients and 109.82 ± 4.61 in the group of longer-treated patients; *p* = 0.701.

## 6. Discussion

Optic neuritis is a rare neurological manifestation in pSS. It may appear as an isolated syndrome or as part of NMOSD. In most available descriptions, optic neuritis appears suddenly, bilaterally, simultaneously, or in a sequential way. It may anticipate clinical pSS symptoms in the time of several years [5,6,20,21]. Etiopathogenesis of optic neuritis in patients with pSS has not been fully explained. Some researchers suggest an ischemic mechanism in the duration of vasculitis in the optic nerve; others point to the action of immunological complexes with the presence of inflammatory infiltrations [21].

A much heavier duration characterizes retrobulbar optic neuritis in NMOSD in pSS than in patients with multiple sclerosis [22,23,24,25]. When untreated, it causes permanent loss of sight. The neuritis may concern one or both optic nerves, and if so, the ailment symptoms in the other eye appear in a short period from the involvement of the first eye.

We did not confirm optic disc fading in our group of patients, nor any significant loss of eyesight. However, essential prolongation of P100 implicit time in one eye only may point to subclinical optic nerve lesion. We observed such a one-time prolongation of latency in only one patient. In the remaining four cases, we recorded abnormally long latencies VEP in both eyes. It is more probable that bilateral changes in VEP result from the involvement of the different parts of the visual pathway. Xing et al. [26] examined pSS patients with performed functional magnetic resonance imaging (FMRI). The authors showed the abnormal brain activity in the visual cortex and frontoparietal junction area in pSS patients, suggesting pathological neuronal dysfunction in these regions. Additionally, Zhang [11] showed the alterations in hippocampal functional connectivity in pSS by using resting-state functional magnetic resonance imaging (rs-fMRI). The white matter hyperintensity score negatively correlated with the functional connectivity between the left hippocampus and right inferior occipital gray/inferior temporal gray.

There is a diversity of findings from other studies on VEP in pSS patients, with only 12% of abnormalities reported by Hietaharju et al. [27] and more than 60% found by Delalande et al. [10]. Hietaharju et al. analyzed a group of 48 patients with Sjögren’s Syndrome, 56% of whom had neurological disturbances; the most common manifestations were entrapment neuropathies in 19% of patients and polyneuropathy in 15% of patients. In the group studied by Delalande et al., despite a small percentage of optic neuritis, more than 40% of patients presented with focal or multifocal brain involvement, which has probably affected the further part of the visual pathway. In the presented study, we did not find any structural changes in imaging studies of the brain, but changes in the activity of the bioelectric visual pathway measured by VEP were demonstrated. The authors of the current article emphasize the importance of VEP in detecting functional changes, especially in the early, subclinical stage of the disease.

Lesions to the optic pathway in pSS may show vascular or, less often, inflammatory-demyelinating character [28,29]. It is estimated that 60% of pSS patients exhibit anti-SSA antibodies and 40% anti-SSB antibodies. Anti-Ro (SS-A) antibodies may play an important part in vascular lesions in the central nervous system. In patients with these antibodies, nervous system involvement occurs more often. Based on head scan clinic pictures (computer tomography, magnetic resonance, angiography of brain vessels), more extensive changes are present than in patients without these antibodies [10]. We have not found any relation between incorrect VEP parameters and anti-SSA/SSB antibodies.

As to predictive factors of nervous system engagement in pSS, primary cryoglobulinemia is one; another is a higher activity of disease with vasculitis that requires aggressive immunosuppressive treatment. Our study confirmed the influence of Ro52 antibodies on VEP parameters: in patients with Ro52 antibodies, substantially statistically prolonged P100 implicit time was observed. The role of anti-Ro52 antibodies in pSS and systemic connective tissue diseases is still incompletely understood. As recent studies have shown, these antibodies are more common in patients with lung involvement [30]. Recent scientific reports emphasize that testing for autoantibodies against anti-Ro52/anti-Ro60 peptides may guide diagnosis, classify clinical manifestations in disease entities and define prognosis in certain autoimmune disorders. A distinct weight could be given to the isolated anti-Ro specificities in the SS classification criteria [31,32,33].

It is notable that VEP lesions did not correlate with the lowered value of complement components C3 and C4, anti-SSA/SSB antibodies, and positive FS (≥1), which may prove the higher activity of the disease process. According to some research, the relation between the presence of palpable purpura that may prove vasculitis was not confirmed, and this mechanism of vasculitis, according to some research, plays a pivotal role in neuron damage in pSS.

In patients with aching joints, shorter P100 implicit time (*p* = 0.03) was shown compared to patients without this symptom. The P100 implicit time prolongation denotes the impairment of the central nervous system, and may be connected with cortex hyperexcitability, which was shown, e.g., on animal models [34]. Different results in the correlation value of wave P10 SEP (peripheral damage) and wave P100 VEP (central damage) may stem from a distinct reaction to pain in the peripheral and central parts of the nervous system. Prolongation of P100 implicit time VEP in the study group may be associated not only with structural lesions in the visual pathway area, but also with the activity change of neurotransmitters, e.g., with the gamma amino butyric acid (GABA) receptor function which modulates N1-P2 responds evoked by standard flash and registered from the primary visual cortex. Bale et al. [35] studied Agonists and antagonists at the *N*-methyl-D-aspartate (NMDA), GABA receptors, and nicotinic acetylcholine receptors on eye evoked potentials. At luminal stimulation with 4.55 Hz frequency, after serving medications, the cited researchers assessed the steady-state amplitude VEP at 1× stimulus frequency (F1) and 2× stimulus frequency (F2). The results point to the role of glutamate and nicotinic acetylcholine receptors at F1 and F2 stimulation, while GABA receptors contribute to F1. It was suggested that hyperpolarizing and GABA inhibitors play a significant role in regulating “critical periods” plasticity in the visual cortex.

We confirmed essentially higher amplitude of wave P100/N45 in the study group. High P100-N145 amplitude may show cerebral hyperexcitability in patients with pSS and may also be explained by the hypothesis of excitotoxic damage to neurons [36,37].

The limitation of the presented study was a relatively small group of patients with pSS and short observation time. As the majority of patients were being treated with hydroxychloroquine/chloroquine (and underwent regular ophthalmological control), it was not possible to exclude reliably the impact of medication upon VEP parameters.

In monitoring potential HCQ toxicity in the patients treated for various connective tissue diseases (Sjogren’s Syndrome, rheumatoid arthritis, lupus erythematosus, and juvenile idiopathic arthritis), screening ophtalmological examination is performed, with addition of electroretinography or spectral-domain optical coherence tomography, when necessary [38,39]. In the available literature, there is no evidence of using VEP in this field. Therefore, it seems feasible to further investigate the usefulness of VEP for this purpose, considering that this method is non-invasive, repeatable, and low-cost. The idea of using VEP for monitoring potential HCQ retinal toxicity seems promising and deserves further investigation, which we plan to undertake.

## 7. Conclusions

pSS patients without CNS involvement presented with dysfunction of visual pathway, as revealed by VEP abnormalities. Relationships were found between VEP parameters and with the presence of anti-Ro52 antibodies and aching joints. VEP may be a useful method for assessment and monitoring of subclinical visual deficit in the course of pSS.

## Figures and Tables

**Figure 1 jcm-10-04196-f001:**
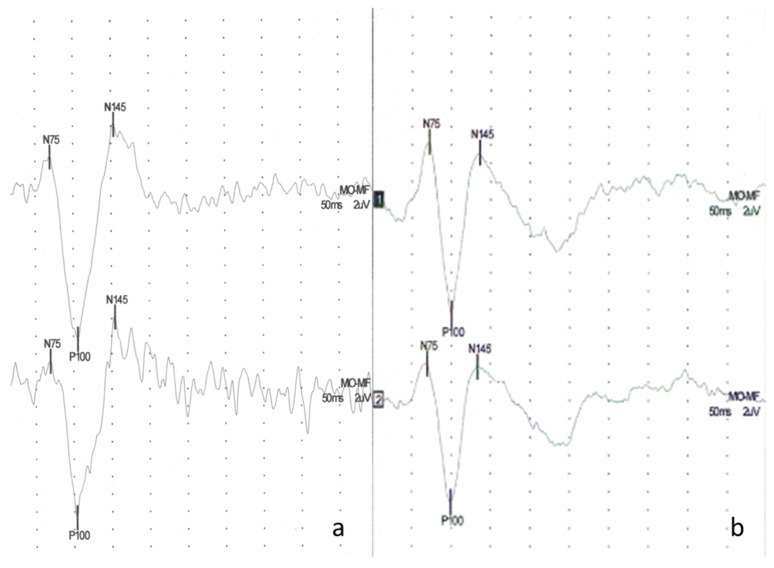
Example of visual evoked potentials (VEP) (**a**) in a patient with pSS—bilaterally normal latency of response P100 (left side 105 ms, right side 103 ms), high P100-N145 amplitude (left side 24.7 µV, right side Po 23.2 µV), (**b**) a typical VEP waveform of the control subject bilaterally normal latency of response P100 (left side 100 ms, right side 98.5 ms) and P100-N145 amplitude (left side 17.7 uV, right side 15.0 uV).

**Figure 2 jcm-10-04196-f002:**
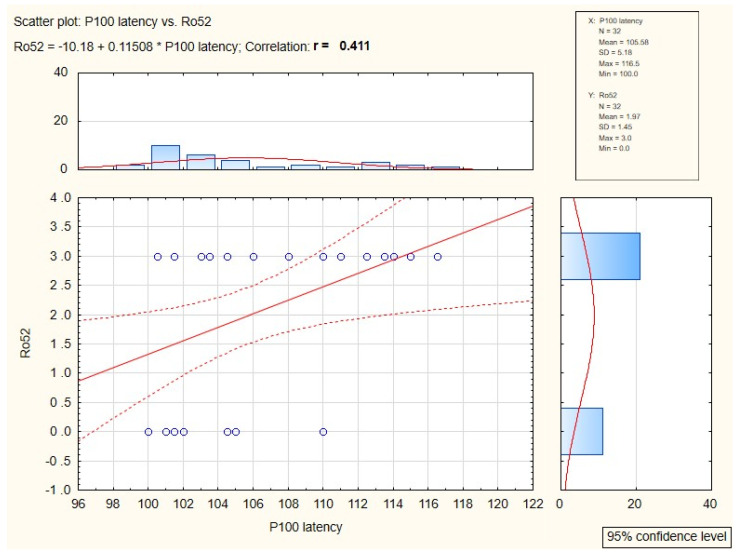
Diagram of correlation between anti-Ro52 antibodies and P100 implicit time value.

**Figure 3 jcm-10-04196-f003:**
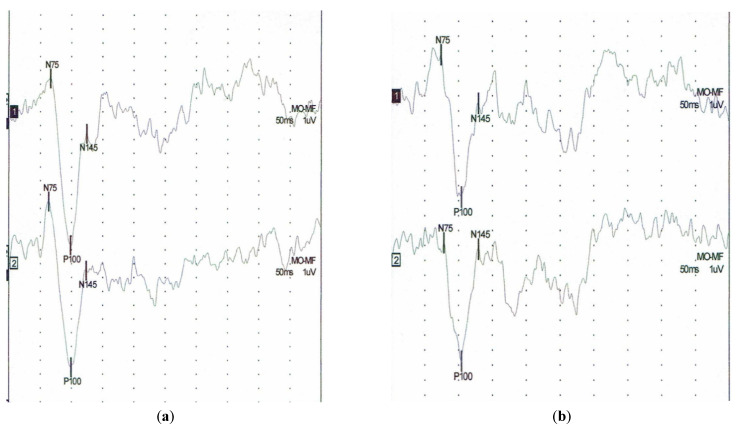
Example of visual evoked potentials (VEP) in the patient with pSS (**a**) baseline study latency of P100 (left side 98 ms, right side 98.5 ms), (**b**) follow up after 6 years latency of P100 (left side 104 ms, right side 104 ms).

**Table 1 jcm-10-04196-t001:** The baseline characteristics of patients with primary Sjögren syndrome (pSS).

Age (Median) at Time of pSS Diagnosis (Years)	50 (35–68)		
Number of patients	Females		31
	Males		1
Comorbidities (n)	Lipid disorder		1
	Cholelithiasis		1
	Hypertension (controlled)		3
	Thyroid nodules		9
Clinical manifestations of pSS at the moment of examination, n (%)	glandular	Xeroophtalmia	32 (100%)
		Xerostomia	31 (97%)
		Parotid gland enlargement	14 (44%)
	extraglandular	Skin lesions	10 (31%)
		Respiratory system involvement	19 (59%)
		Peripheral arthritis	28 (87%)
Sialadenitis with a focus score ≥ 1, n (%)	29 (90%)		
Autoantibodies, n (%)	ANA		26 (81%)
	Ro52-antibody positive		22 (69%)
	SSA-antibody positive		26 (81%)
	SSB-antibody positive		21 (66%)

**Table 2 jcm-10-04196-t002:** Visual evoked potentials in the study group and in the control group.

VEP		Study Group *n* = 32			Control Group *n* = 50			*p*-Value
		Median	Mean	SD	Median	Mean	SD	
Latency (ms)	N75	71.0	73.2	7.6	69.0	70.4	4.6	0.072
	P100	103.5	105.5	5.1	99.50	100.5	3.9	0.000
	N145	143	145.7	14.2	146	140.8	10.9	0.068
Amplitude (μV)	P100-N145	11.2	12.3	4.1	9.98	9.4	3.0	0.000

**Table 3 jcm-10-04196-t003:** Visual evoked potentials in the study group after 6 years.

VEP		Mean ± SD
Latency (ms)	N75	77.9 ± 7.33
	P100	109.37 ± 5.67
	N145	146.80 ± 12.19
Amplitude (μV)	P100-N145	9.89 ± 3.38

**Table 4 jcm-10-04196-t004:** Visual evoked potentials in the study group during baseline and in follow up after 6 years.

VEP		Study Group at Baseline *n* = 32			Study Group in Follow up after 6 Years *n* = 32			*p*-Value
		Median	Mean	SD	Median	Mean	SD	
Latency (ms)	N75	71.0	73.2	7.6	77.0	77.9	7.33	0.001
	P100	103.5	105.5	5.1	108.75	109.37	5.67	0.002
	N145	143	145.7	14.2	149.25	146.8	12.19	0.751
Amplitude (μV)	P100-N145	11.2	12.3	4.1	9.83	9.89	3.38	0.014

## Data Availability

The data presented in this study are available on request from the corresponding author.

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
