# Peer review of "Visual Evoked Potentials as Potential Biomarkers of Visual Function in Patients with Primary Sjögren’s Syndrome"

_jcm, 2021, doi:10.3390/jcm10184196_

Round 1
Reviewer 1 Report
Dziadkowiak et al investigate the function of the visual pathway in patients with primary Sjögren’s syndrome (pSS) without focal symptoms of central nervous system disorder. The function of the visual pathway was examined by assessment of visual evoked potentials (VEP) at two-time points; baseline and after 6 years and was compared to those of healthy individuals. pSS patients were found to suffer from dysfunction of the visual pathway as this assessed by prolonged P100 implicit time compared to healthy, while VEPs were also increased 6 years after diagnosis compared to baseline. Abnormal VEP at baseline correlated with anti-Ro52 34 antibodies and aching joints. These findings are interesting; however, the following issues need attention.
- The only control group used is healthy individuals. Similar studies should be also performed in patients with another autoimmune disease; age and sex-matched and under similar medication.
- How many of the patients had vasculitic involvement (vasculitis, purpura, etc)? Does this correlate with VEP abnormalities? Do the VEPs correlate with the extend of tear production, as this estimated by Schirmer’s test?
- Previous published literature reporting abnormal VEP in a percentage of pSS patients are not acknowledged and discussed. These studies find lower percentage of pSS patients with abnormal VEP findings (DOI: 10.1097/01.md.0000141099.53742.16 & DOI: 10.1111/j.1600-0404.1990.tb00951.x). How the authors explain this discrepancy. This should be also included in the manuscript.
Author Response
Reviewer 1
Dziadkowiak et al investigate the function of the visual pathway in patients with primary Sjögren’s syndrome (pSS) without focal symptoms of central nervous system disorder. The function of the visual pathway was examined by assessment of visual evoked potentials (VEP) at two-time points; baseline and after 6 years and was compared to those of healthy individuals. pSS patients were found to suffer from dysfunction of the visual pathway as this assessed by prolonged P100 implicit time compared to healthy, while VEPs were also increased 6 years after diagnosis compared to baseline. Abnormal VEP at baseline correlated with anti-Ro52 34 antibodies and aching joints. These findings are interesting; however, the following issues need attention.
- The only control group used is healthy individuals. Similar studies should be also performed in patients with another autoimmune disease; age and sex-matched and under similar medication.
Further planned investigations would include larger group of pSS patients as well as comparative groups of subjects with other connective tissue diseases (e.g. systemic lupus erythematosus), and would focus on effect of CNS involvement and particular therapeutic agents (especially with regard to chloroquine) upon parameters of evoked potentials.
- How many of the patients had vasculitic involvement (vasculitis, purpura, etc)? Does this correlate with VEP abnormalities? Do the VEPs correlate with the extend of tear production, as this estimated by Schirmer’s test?
Skin lesions were found in 10 of patients, of which 6 had a palpable pulpura (i.e. lesions of the vasculitis type) located on the lower limbs. In 96% of patients, the Schirmer's test value was less than or equal to 5 mm / 5 min.
- Previous published literature reporting abnormal VEP in a percentage of pSS patients are not acknowledged and discussed. These studies find lower percentage of pSS patients with abnormal VEP findings (DOI: 10.1097/01.md.0000141099.53742.16 & DOI: 10.1111/j.1600-0404.1990.tb00951.x). How the authors explain this discrepancy. This should be also included in the manuscript.
The references were cited and their results discussed.
Reviewer 2 Report
- In the introduction, the significance of VEP is not very clear. Why measuring VEP matter for pSS patients?
- The method for Ro52 measurement is not appropriate, there were commercially available assay to check Ro52 in quantitative/semi-quantitative way. In current paper, the Ro52 is either 0 or 3.
- In the introduction, the author mention "Eye disorders in patients with pSS may also occur in the course of chloroquine treatment.", and in the current cohorts, "In the study group, 60% of patients have treated with hydroxychloroquine in the average dose 200 mg per day, 25% with chloroquine in the dose 250 mg per twenty-four hours. What's the impact of this treatment to VEP? there is no discussion.
Author Response
Reviewer 2
- In the introduction, the significance of VEP is not very clear. Why measuring VEP matter for pSS patients?
The role of VEP in early detection of visual involvement and its clinical significance was
highlighted.
- The method for Ro52 measurement is not appropriate, there were commercially available assay to check Ro52 in quantitative/semi-quantitative way. In current paper, the Ro52 is either 0 or 3.
Anti-Ro52, anti-SSA and anti-SSB antibodies determined by ELISA method. The anti-Ro52 level was assessed relying on the intensity of the color directly linked to the number of antibodies in the patient sample (0- no antibodies, 3 - the most intense colour).
- In the introduction, the author mention "Eye disorders in patients with pSS may also occur in the course of chloroquine treatment.", and in the current cohorts, "In the study group, 60% of patients have treated with hydroxychloroquine in the average dose 200 mg per day, 25% with chloroquine in the dose 250 mg per twenty-four hours. What's the impact of this treatment to VEP? there is no discussion.
In the study group, the possibility of side effects in the course of chloroquine treatment was taken into account, therefore all patients underwent a normal ophthalmological examination.Because the majority of patients were being treated with (hydroxy)chloroquine (and underwent regular ophthalmological control) , it was not possible to exclude reliably the impact of medication upon VEP parameters. Analysis of this issue is planned as the subject of our future investigations.
Round 2
Reviewer 1 Report
The revised version of the manuscript still does not address the following issues:
- The vasculitic involvement correlates with VEP abnormalities?
- 96% of the patients studied had positive Schirmer’s test (≤5mm/ 5 min). Does the extend of tear production (number of mm in Schirmer’s test) associates with VEP abnormalities?
Author Response
Point 1: The vasculitic involvement correlates with VEP abnormalities?
Response 1: Skin changes in the course of vasculitis did not correlate with changes in VEP. In the text of the article, we wrote about it in the summary of the parameters that did not correlate with the VEP parameters: „Statistical significance between VEP parameters and the degree of xerophthalmia, the disease's duration, and the presence of skin lesions characteristic for vasculitis, lesions in major salivary glands were not proved.”
Point 2: 96% of the patients studied had positive Schirmer’s test (≤5mm/ 5 min). Does the extend of tear production (number of mm in Schirmer’s test) associates with VEP abnormalities?
Response 2: The Schirmer’s test was not performed on the same day as the VEP. According to our statistician, due to the too large number of patients with a positive Schirmer's test, it was not possible to statistically calculate the relationship between the subgroups with a negative and a positive test.